# Water Wins, Communication Matters: School-Based Intervention to Reduce Intake of Sugar-Sweetened Beverages and Increase Intake of Water

**DOI:** 10.3390/nu14071346

**Published:** 2022-03-23

**Authors:** Tanja Kamin, Barbara Koroušić Seljak, Nataša Fidler Mis

**Affiliations:** 1Faculty of Social Sciences, University of Ljubljana, SI-1000 Ljubljana, Slovenia; tanja.kamin@fdv.uni-lj.si; 2Computer Systems Department, Jožef Stefan Institute, SI-1000 Ljubljana, Slovenia; barbara.korousic@ijs.si; 3Department of Gastroenterology, Hepatology and Nutrition, University Children’s Hospital, University Medical Centre Ljubljana, SI-1000 Ljubljana, Slovenia

**Keywords:** social marketing, health communication, childhood obesity, water, sugar-sweetened beverages

## Abstract

We compared three interventions designed for reducing the consumption of sugar-sweetened beverages (SSBs) aimed at decreasing the risk of overweight and obesity among children. We included three experimental (n = 508) and one control school (n = 164) in Slovenia (672 children; 10–16 years) to evaluate interventions that influence behaviour change via environmental (E), communication (C), or combined (i.e., double) environmental and communication approaches (EC) compared to no intervention (NOI). Data of children from the ‘intervention’ and ‘non-intervention’ schools were compared before and after the interventions. The quantity of water consumed (average, mL/day) by children increased in the C and EC schools, while it decreased in the E and NOI schools. Children in the C and EC schools consumed less beverages with sugar (SSBs + fruit juices), and sweet beverages (beverages with: sugar, low-calorie and/or noncaloric sweeteners) but consumed more juices. The awareness about the health risks of SSB consumption improved among children of the ‘combined intervention’ EC school and was significantly different from the awareness among children of other schools (*p* = 0.03). A communication intervention in the school environment has more potential to reduce the intake of SSBs than a sole environmental intervention, but optimum results can be obtained when combined with environmental changes.

## 1. Introduction

Childhood obesity is one of the most serious health challenges of the 21st century. The number of obese children and adolescents (aged 5–19 years) worldwide rose tenfold from 1975 to 2016 [1]. One of the reasons is related to the current food environment, which is characterised by a cheap and abundant free sugar supply in solid and liquid forms. Several systematic reviews and meta-analyses of prospective cohort studies and randomised controlled trials provide evidence that the consumption of sugar-sweetened beverages (SSBs) promotes weight gain [2,3,4,5]. In the Health Behaviour in School-aged Children (HBSC) study, Slovenian 15-year-olds were at the top of 41 countries in Europe and North America in terms of the frequency of adolescents drinking soft drinks daily. Slovenian 11-year-olds were in second place, whereas 13-year-olds were in third place. The highest frequency of SSBs consumption on a daily basis in Slovenia was reported in 15-year-olds (boys and girls: 49% and 39%), followed by 13-year-olds (41% and 31%) and 11-year-olds (36% and 27%), with the prevalence being higher in boys [6]. Studies have clearly shown that reduced consumption of carbonated beverages (SSBs) or reduced consumption of SSBs on behalf of their replacement with sugar-free, sucralose-sweetened beverages leads to a reduction in the number of overweight and obese children [7,8]. These studies have highlighted the benefits of reducing SSB consumption as part of an active intervention programme for childhood obesity; however, they point to the need for continuing intervention to promote a healthy food environment and healthy behaviours in children to maintain the effect [9,10].

Thus, researchers and public health professionals focusing on obesity prevention have been encouraged to pursue their efforts in finding effective ways to promote water drinking and reduce SSB consumption among children [11,12,13,14]. A recent systematic literature review and meta-analysis of interventions to reduce the consumption of SSBs and increase water intake among children showed significant decreases in SSB consumption among children for community and school-based studies and significantly increased water intake in home settings [15]. The school is an appealing site for designing interventions, given that children and adolescents spend the largest amount of time in a day in school, and that they eat at least one meal per day in school [16]. Evidence shows that school-based interventions can reduce the incidence of being overweight [7,8,17,18]. Extensive systematic reviews of the efficacy of school-based interventions aimed at decreasing SSBs among adolescents shows that those interventions successful in changing behaviour at least in a short time, are either aimed to influence behaviour via a communicational approach or via environmental changes (e.g., removing SSBs from school canteen). Only six of the interventions were designed according to an ecological model—targeting both individuals and their environments—and combined communicational and environmental approaches. Vezina-Im et al. [19] aimed to identify the most effective intervention technique for SSB-related behaviour change; however, they were unable to draw unanimous conclusions, given very different study designs and different combinations of behaviourchange techniques in experimental groups. However, they did recommend that there should be more studies simultaneously targeting individuals and their environments [19]. This is in line with the findings of Steyn et al. [20], who reviewed best practices of school-based nutrition interventions (although not particularly focused only on SSB intake) chosen by WHO, and proposed that school-based interventions have more effect on behaviour change and clinical change when trained teachers offered a nutrition-based curriculum at school; intervention were grounded on a firm theory of behaviour, such as social cognitive, social marketing or stages of the change; interventions included a food service component; interventions included parents or families of children involved in the intervention; and interventions also included a physical activity programme. Successful school-based interventions that addressed both environment and individual stressed the importance of engaging youth cognitively and emotionally in the intervention and providing changes in the environment that would support the formation of new, healthier habits [11,21]. Several studies suggest that the environments in schools can easily be made more supportive of recommended behaviour by limiting access to competitive foods and exposing those that support the desired behaviour [11,12,22,23].

However, environmental changes alone might not lead to sustainable behaviour changes [24]. Previous studies shows that communication interventions in the school environment, designed by principles of social marketing, which holds behavioural change as its bottom line, is essentially target-group driven and emphasises the co-creation of attractive exchanges that encourage social change [25,26], has more potential for reduced intake of SSBs than sole environmental intervention in the school environment, but receives the best results when combined with environmental changes [11,21].

In this study, we aimed to develop and test interventions that educate (increase health literacy, which according to Zarcadoolas et al. [27] refers to skills and competencies that people need to comprehend, evaluate, and use health information and concepts to make informed choices, reduce health risks, and increase quality of life), enable (make healthier alternatives more easily available), and engage (making healthier alternatives more appealing than unhealthy choices) target audiences (primary-school children, their parents and relatives, and teachers and other school co-workers) in selected primary schools.

Thus, the objective of this study was to compare and assess the effects of different school-based interventions, which were informed by formative research and developed with a social marketing user-centred approach aimed at reducing SSB intake and increasing water consumption among primary school children. With a quasi-experimental design, we evaluated three types of interventions: those that influence behaviour change (i) via an environmental (E) approach; (ii) via a communication (C) approach; and (iii) those that combine environmental and communication approaches (EC); versus (iv) no intervention (NOI).

## 2. Materials and Methods

### 2.1. Intervention

The emphasis of the interventions was on replacing SSBs with non-caloric beverages in the school environment as a strategy to decrease SSB consumption. Tap water is an inexpensive option for providing hydration to children in school settings, but it needs to be made appealing to make it a preferable choice. We chose a social marketing approach in designing the interventions, since evidence demonstrates that well-employed social marketing benchmark criteria—*behavioural objectives*, *segmentation*, *formative research*, *exchange*, *marketing mix*, *and competition* [25]—offer great potential in behaviour and social change [28,29,30].

Our principle *behavioural objective* was to decrease SSB consumption among children but also to change awareness about the health hazards related to the consumption of SSBs. We conducted a literature review on eating habits among primary school youth in Slovenia [3]; however, according to the user-centred approach, we needed to capture more insights into the segment of children that we intended to address in our study. Thus, we conducted *formative research* in selected primary schools. We conducted (i) focus groups with primary-school-age youths (i.e., grades 6–9 in Slovenia, i.e., aged 10–16 years) addressing their attitudes and beliefs regarding thirst, hydration, types of beverages, and their choices for releasing thirst; and (ii) interviews with school staff regarding hydration in school, their perceptions of offer and demand with regard to the beverages available in the school environment. The strongest takeaway message from the focus groups with children was that health discourse in promoting drinking water cannot compete with appealing messages and packaging of SSBs’ brands. Based on the literature review and formative research, we identified one primary and two secondary *target groups* for our study and developed the intervention strategy according to their specifics (Table 1). *Competition* of commercial entities that promote beverages, among which those with added sugar prevail, mostly uses appeals of happiness, entertainment, and pleasure, often combined with funny cartoon characters, famous people from the entertainment or sports industry when competing for the hearts and minds of young consumers. Children often ignore and disregard messages from health authorities about the health hazards of SSBs [31]. As other scholars [14,26], we also acknowledged the importance of branding the intervention to children when competing for pupil’s attention in promoting consumption of water and non-caloric beverages. We aimed to design an attractive *exchange* situation in which children would recognize the added value of water consumption, give up drinking SSBs and adopt drinking water, and recognise the personal benefits of their behaviour change, among which the coolness and smartness of drinking water was exposed. We employed several elements of *marketing mix*, focusing on activities in the selected schools, making healthy choices for hydration (water) more easily available, and encouraging engagement of school children in the programme with several communication activities that we *branded* under the slogan “Water wins!”

Interventions developed in this way included both communication and an environmental approach. We aimed to test the strength of both approaches in case they worked together or separately. Thus, we tested three types of interventions: (i) a communication-based approach (directed to three target groups: children, parents, school staff) in school C; (ii) an approach based on change of environment (directed to two target groups: children, school staff) in school E; and (iii) an intervention that included both (directed to three target groups: children, parents, school staff) in school EC. In Table 1 we briefly describe communication and environmental interventions, according to the chosen target groups (who the intervention was for), communication goal of the intervention (educate, engage, and/or enable), lead communicators of the messages, and, in the case of communication intervention, also the tone and appeal of the communication. Both communication and environmental intervention were designed for and were part of the study, implemented in three primary schools (C, EC and E) in Ljubljana. The study included another primary school, in which there was no intervention carried out (NOI). The quasi-experimental design of the study is described in the next section (Section 2.2). 

### 2.2. Evaluation Study Design and Setting

School-based interventions with a quasi-experimental design were conducted in four primary schools in Ljubljana, the capital of Slovenia, and its surroundings, from September 2012 through June 2013. The study involved 672 children from four primary schools, aged 10 to 16 years. There were slightly more girls (53.7%) than boys in the final sample (Table 2).

Active (five months) and passive (four months) interventions (see details in Section 2.4. Data collection and Table 3) were performed in three schools (C, E, EC), while non-intervention was performed in one (control) school (NOI). The active interventions included communication and environmental interventions. Communication interventions aimed to influence attitudes and behaviour by informing children, their parents, and teachers about the positive health benefits of drinking water and the negative consequences of drinking SSBs while encouraging healthy drinking choices. Environmental interventions refer to a changed supply of drinks in schools from SSBs to water, carbonated water, and/or unsweetened fruit or herbal teas.

Data collected for children of the ‘intervention’ schools (n = 508) was compared with data collected for the ‘non-intervention group of children’ (n = 164) before and after the interventions. There were no eligibility criteria for including children in the study except for age and willingness to participate; hence, participation was determined at the school level.

Regarding the context of the interventions, it should be noted that none of the interventions explicitly focused on SSBs were designed and implemented in Slovenia prior to our study. However, general campaigns for healthy eating and regular physical activity among school children and the general population have been quite common [32]. An important measure for reducing SSBs’ consumption in schools was taken in 2010, two years prior to our study, when Slovenia adopted a ban on vending machines on all primary school premises [33]. 

### 2.3. Ethics Approval

Informed consent procedures were followed for all children. Parents (or guardians) gave their informed written consent and permission for their child to participate in the study; verbal assent was obtained from each child. The study protocol was approved by the National Medical Ethics Committee of the Republic of Slovenia (approval document 47/09/12).

### 2.4. Data Collection

At the beginning of the study (i.e., before the interventions in September 2012) and after the active interventions ended (in February 2013), the children anonymously completed two questionnaires: (i) a questionnaire about personal data, drinking habits, and anthropometry (Q1); and (ii) a questionnaire about the awareness of negative consequences of drinking SSBs (Q2) (Table 3). 

Each child was assigned an anonymous study code. The questionnaires were implemented in an electronic form as part of the Open Platform for Clinical Nutrition [34,35].

Results of the first questionnaire (Q1) about drinking habits and anthropometry performed at the beginning of the study and after the active interventions were collected from 465 children (69.2%). These results provided us with participants’ (i) personal data about sex, age, school, and class; and (ii) data about drinking habits. The questionnaire included sixteen questions grouped into eight sections. Each question had eight possible answers, which provided information about the frequency of drinking (from ‘never’ through ‘four and more times per day’). For each positive answer, the participants provided the number of drinking units, where one glass was equivalent to 200 mL.

There were 340 (65.0%) children who completed the second questionnaire (Q2) about their awareness of the negative consequences of drinking SSBs, which was also administered before and after the interventions. The questionnaire had nine questions with five possible answers (from fully agree to disagree) and six questions with alternative answers, from which only one that best reflected the participant’s behaviour in the previous week could be selected. Q2 included photos of different kinds of drinks, and with each drink, a glass with 200 mL of drink in it, to ease answering.

The anthropometric measurements were carried out twice (i.e., before the interventions in September 2012 and at the end of the study in June 2013) by physical education teachers and were collected from 541 children (88.4%). We excluded those that, in the second measurement, were more than 15 cm shorter or more than 15 cm taller, had a body mass of more than 10 kg higher or more than 15 kg lower (as all this could only be due to measurement errors), or lacked anthropometric measurements.

### 2.5. Analysis

The first analysis compared the characteristics of the children who were excluded from the study with the characteristics of the children involved. We applied the Mann–Whitney signed-rank test as an alternative to the *t*-test for independent samples to assess whether, at the beginning of the study, the quantity and the percentage of consumed drinks differed between the involved and excluded participants. The Mann–Whitney signed-rank test was also applied to check the awareness of the negative consequences of drinking SSBs between both groups of participants. To determine whether there were statistically significant differences in the body mass index (BMI) and BMI *z*-scores of the included and excluded children at the first measurement, the Mann–Whitney test was applied. 

In the second analysis, we studied the results of Q1 to determine the self-reported consumption of different types of beverages and the share of each type of consumed beverage among all drinks. Given that normality could not be confirmed, we applied the Wilcoxon signed-rank test as an alternative to the *t*-test for dependent samples to determine whether the self-reported quantity of consumed SSBs decreased after the interventions. To study the differences between the schools, we used the Mann–Whitney test. To determine the effect of active interventions on the proportion of overweight and obese children, we applied a chi-square test. A correlation between BMI and drunk beverages was explored using Spearman’s rank correlation coefficient (*rho*). We used logistic regression as a statistical method for analysing how the proportion of overweight and obese children changed in the ‘intervention’ schools in comparison with the ‘control’ school.

In the third analysis, we studied the results of Q2 to determine an ‘awareness’ index before and after the interventions. Using the Mann–Whitney test, we explored how awareness changed in the school where the double intervention was applied compared to the ‘one intervention’ schools and the ‘control’ school.

The final analysis studied the results of the anthropometric measurements to identify a correlation between the consumption of SSBs and BMI and BMI *z*-scores, as well as the impact of the active interventions on decreasing BMI and BMI *z*-scores (using Spearman’s rank-order correlation and logistic regression).

## 3. Results

### 3.1. Change in Drinking Habits

After the interventions, the quantity of water (mL/day) consumed on average by children of the *C* school and the *EC* school increased, while it decreased in the *NOI* school and the *E* school (Figure 1). After the interventions, children of the *C* school and the *EC* school consumed less SSBs, more juices, less beverages with sugar (SSBs + fruit juices), and less sweet beverages (beverages with: sugar, low-calorie and/or noncaloric sweeteners, such as the sugar alcohols aspartame, acesulfame-K, saccharin, sucralose or stevia). There was only a difference in the quantity of beverages with sweeteners consumed: while this quantity increased among children of the *C* school, it decreased among children of the *EC* school; however, the amounts were minor in all schools. It is interesting that children of the *E* school drank more beverages of all other types than water.

Among all types of drinks across all schools, the quantity of consumed water was the most affected. Considering the results of the Wilcoxon signed-rank test, we can conclude that children of the schools where at least one intervention was performed consumed statistically significantly more water (Z = −2.66, *p* < 0.01); less beverages with added sugar (SSBs) (Z = −3.03, *p* < 0.01); less beverages with sugar (Z = −2.84, *p* < 0.01); and less sweet beverages (Z = −2.66, *p* < 0.01). The quantity of juices and drinks with sweeteners did not statistically significantly differ between groups (*p* > 0.05). The Wilcoxon test was applied because we could not prove the normality of the data set, considering the results of the normality tests (skewness and kurtosis, Shapiro–Wilk). The only exception was water, for which the measures of skewness and kurtosis were −0.01 and 0.94, respectively.

The Mann–Whitney test showed that there was a statistically significantly larger change in the quantity of SSBs consumed by children of the *EC* school compared to other schools (U = 22,746; *p* < 0.01). The same applied for beverages with sugar (SSBs + fruit juices) (U = 22,522; *p* < 0.01) and beverages with added sugar (SSBs) (U = 22,671; *p* < 0.01). The Mann–Whitney test was applied because we could not prove the normality of the data set considering the results of the normality tests (skewness and kurtosis, Shapiro–Wilk).

### 3.2. Awareness of Health Risks Related to the Consumption of SSBs

The awareness of health risks related to consumption of SSB was better after the intervention among children of the *EC* school; it statistically significantly differed from the awareness among children of other schools (U = 11,272, *p* = 0.03). The Mann–Whitney test was applied because we could not prove the normality of the data set considering the results of the normality tests (skewness and kurtosis, Shapiro–Wilk), although Levene’s test proved the assumption that variances of the populations from which the different samples were drawn were equal (F = 2.31, *p* = 0.13).

### 3.3. Risk of Obesity 

The study included more children with normal weight at the start of the study (Table 4); however, the number of children with normal weight at the end of the study increased, while the number of overweight and obese children slightly decreased.

However, there was a very low correlation (Spearman’s rank correlation coefficients in the range from −0.17 to 0.16) between BMI and BMI z-scores of children and the quantity and distribution of drinks consumed across the beverage groups at the end of the study (Table 5).

To assess whether the risk of obesity statistically significantly decreased in the ‘intervention’ schools compared to the *NOI* school, we performed logical regression (Table 6). We found out that in none of the three ‘intervention’ schools was the risk of obesity statistically significantly reduced compared to the *NOI* school.

## 4. Discussion

Our study addresses one of the research gaps in comparing the effects of different combinations of behaviour change techniques in experimental groups under the umbrella of one study design with regard to school children’s SSBs consumption [19]. We designed the interventions in accordance with a social marketing approach and in line with the recommendations of Steyn et al. [20]. As in some other studies [11,21], our findings also stress the importance of developing school-based interventions, which address both environment and individual, engage youth cognitively and emotionally in the intervention, and provide changes in environment that would support forming new, healthier habits.

Our findings show that communication intervention in the school environment, designed by principles of social marketing, showed more potential to reduce the intake of SSBs than a sole environmental intervention in the school environment; however, the best results were obtained when combined with environmental changes. In recent years, interventions in the environment, which influence the way in which choices are made, also called nudging, aiming to shape behaviour in a desired direction [37], have become highly praised. However, it has also been acknowledged that such environmental interventions nudge people to behaviour decisions, which are often automatic, unconscious, and might not transfer into environments that do not nudge people in that same behaviour [38]. These strategies are often based on the assumption—derived from behavioural economics—that people seldom make rational decisions and deliberate choices in their everyday behaviour, especially when this is part of their habits [37].

According to socio-cognitive theory [39], human behaviour is complex, shaped by individual factors but also infrastructure, environmental circumstances, social norms, desires, needs, and other factors that environmental interventions by itself cannot entirely address. Thus, environmental interventions with nudging tools should be seen more as a complement to other interventional approaches, such as communication, than as their substitute [24,38]. Communication intervention, when executed correctly, seems to have a better spill-over effect than environmental intervention alone, because it might also affect drinking habits at home and other non-school environments. Awareness of health risks related to the consumption of SSBs improved the most among children from the double intervention school (EC), hinting that a combination of information and practice leads to the best results, which is consistent with the cultural capital theory that supports the idea that knowledge and practice feed on each other [40].

One of the advantages of our study was that we aimed to measure the effects of interventions by detecting changes not only in awareness and reported drinking habits but also in the BMI of children included in the study; thus, we observed changes in drinking habits with anthropometry. It should be noted that a great majority of the children included in our study had normal weights. Thus, overweight and obese children were underrepresented in the sample. In the whole sample, the number of children with normal weight at the end of the study increased, while the number of overweight and obese children decreased slightly. However, none of the three interventions resulted in a statistically significant reduction in the risk of obesity compared to the ‘control’ school. This could be related to the very small percentage of overweight and obese children in the sample, but it also indicates how stubborn habits are and how difficult they are to change. However, the clinical part of the study was already performed in 2012 and 2013, and there has been no study since then in this area. Further, the regulations concerning beverage intake in schools have not changed since then.

Our study has several limitations, some of them derived from its quasi-experimental design, which focused on a single dietary behaviour in the school environment with a diverse sample. Although BMI results from various factors, such as other dietary intake and physical activity, we measured only the assessment of SSBs’ consumption, which was self-reported in a survey and might have been subjected to social desirability. The active intervention lasted for only five months, and the data collection did not follow immediately after the interventions. It is therefore possible that the results on risk awareness and changes in water and SSB consumption would have been even more encouraging immediately after the active interventions. A slight increase in the number of normal weight children in the study could support this reasoning. Despite this limitation, collecting data after the active and passive interventions is more likely to accurately reflect the true impact of our study. Another limitation of the study is that the BMI measurements were conducted by school sport teachers, not health professionals. This led to some mistakes in measurements that forced us to exclude some children from the final sample of the study. Further, the decrease in the number of children between the first and the second measurement was mainly on the account of those with higher BMI, who were much more reluctant to be weighted. Other studies have shown [7,9,10] that children with higher BMI usually show better results after active intervention, both in decreased BMI and in changed habits. The results of our study, in addition to previous literature data, show that dealing with perceived weight bias is an important consideration for future school-based interventions.

## 5. Conclusions

Our study contributed to the corpus of studies that aim at reducing consumption of SSBs among primary school children and demonstrating that communication interventions in the school environment, designed by the principles of social marketing, have more potential for a reduced intake of SSBs than sole environmental change in the school environment, but yields better results when combined with environmental changes.

## Figures and Tables

**Figure 1 nutrients-14-01346-f001:**
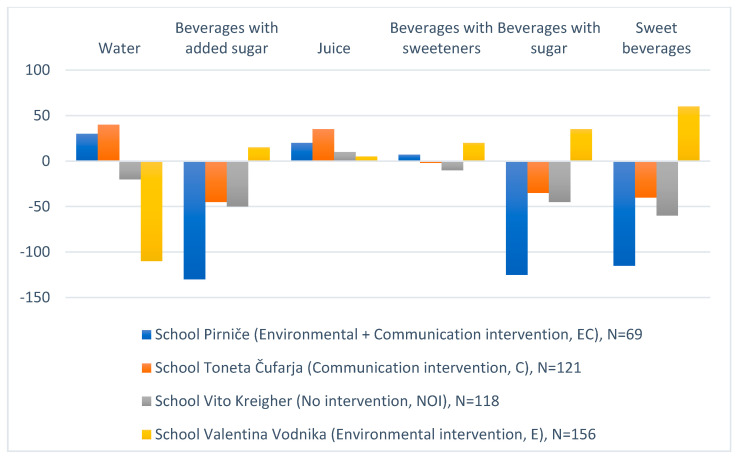
Quantity of beverages (mL/day) consumed on average by children of the communication intervention school (C); the environmental and communication intervention school (EC); the control school (NOI); and the environmental intervention school (E).

**Table 1 nutrients-14-01346-t001:** Interventions according to the chosen target groups.

Target Group	Goal	Communicators for Disseminating Message of the Intervention	Tone of Communication and the Main Message	Appeal	Communication Intervention Channels (15 September 2012–15 February 2013) (In Schools C and EC)	Environmental Intervention (15 September 2012–15 February 2013) (In Schools E and EC)
Primary school children (10–16 years)	Educate Engage Enable	Children (peers)	Humorous, teasing, motivational, informal, exposing fun facts about water and positive effects of water for living beings but not primarily focused on health concern Slogan of the communication: Water wins!	Emotional	Posters Leaflets Web page FB profile Stickers Event “Water day” for children ^wd^ Creative competition with awards	SSBs are: - not on the school menu anymore, - replaced with water ^W^ Environment with offers of water ^w^
Parents	Educate	Nutritionist, health authority	Formal, informational, exposing negative effects of SSBs for children	Rational, cognitive	Lecture Brochure	
School staff (principal, teachers, school meal organiser)	Educate Engage	Nutritious expert, health authority	Formal, informational, exposing: - negative health effects of SSBs for children, - environmental and - economic effects of changing school menus with water ^w^ Presenting useful practical solutions for school environment ^PS^	Rational, cognitive	Personal, face to face lecture Individual face to face discussions	Change in the school meals menus

^W^ water, unsweetened tea, or mineral water; ^wd^ recreation-science day aimed at learning through play; ^PS^ Water bars, water bottles, tap water, personalised cups.

**Table 2 nutrients-14-01346-t002:** Study sample.

School Intervention	Tone Čufar Communication (C)	Valentin Vodnik Environmental ^e^(E) ^e^	Pirniče Environmental ^e^ + Communication (EC)	Vita Kraigherja No Intervention (NOI)/	Together
Girls	77	138	46	100	361
Boys	80	116	51	64	311
Together	157	254	97	164	672

^e^ Water.

**Table 3 nutrients-14-01346-t003:** Data collection according to interventions: active (September 2012–February 2013; 5 months), passive (March 2013–June 2013; 4 months), and total (September 2012–June 2013; 9 months).

	1st Data Collection	2nd Data Collection	N	N
Questionnaire about:				
- Drinking habits (Q1)	September 2012	February 2013	672	465
- The awareness of the negative consequences of drinking SSBs (Q2)	September 2012	February 2013	523	340
Anthropometric measurements	September 2012	June 2013	612	541

**Table 4 nutrients-14-01346-t004:** Anthropometric measures of children before and after the interventions according to the classification of the International Obesity Task Force (IOTF [36]) ^1^.

	BMI	1st Measurement		2nd Measurement	
		N	%	N	%
Underweight	<16	2	0.4	1	0.3
Normal weight	≥16–<17	7	1.4	9	1.7
	≥17–<18.5	27	5,0	31	5.7
	≥18.5–<25	378	69.9	390	72.1
	Total	414	76.5	431	79.7
Overweight Obesity	≥25–<30	108	20.0	93	17.2
	≥30	19	3.51	17	3.14
	Total	127	23.5	110	20.3

BMI, Body Mass Index; ^1^ To define the groups of overweight and obese children, we used the categorisation defined by the IOTF [36]; N, number of children; %, percentage of children.

**Table 5 nutrients-14-01346-t005:** Relationship between BMI and BMI z-scores with the absolute amounts and percentages of different beverages consumed after active and passive intervention (Spearman’s correlation coefficient).

	Quantity of Beverages	Water	Beverages with Added Sugar	Juice	Beverages with Sweeteners	Beverages with Sugar	Sweet Beverages
BMI	mL/day	0.09	−0.11	−0.08	0.05	−0.10	−0.10
	%	0.14	−0.12	−0.09	0.05	−0.14	−0.14
BMI z-score	mL/day	0.13	−0.14	−0.08	0.07	−0.12	−0.11
	%	0.16	−0.17	−0.10	0.07	−0.17	−0.16

BMI, Body Mass Index.

**Table 6 nutrients-14-01346-t006:** Risk of obesity in intervention groups (Communication (C); Environmental (E); and double intervention Communication and Environmental (EC)).

School	Odds Ratio	Confidence Interval	
		Lower Limit	Upper Limit
Tone Čufar (C)	0.09	0.00	2.02
Valentin Vodnik (E)	0.33	0.03	4.29
Pirniče (EC)	0.17	0.01	3.59

## Data Availability

Data supporting reported results can be obtained upon request to natasa.fidler@kclj.si.

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
