# Peer review of "Water Wins, Communication Matters: School-Based Intervention to Reduce Intake of Sugar-Sweetened Beverages and Increase Intake of Water"

_nutrients, 2022, doi:10.3390/nu14071346_

Round 1

Reviewer 1 Report

A comparison between approaches to reducing SSB consumption in a school setting is of interest readers. however, there are revisions that need to be made before this manuscript is ready for publication. A thorough review for spelling and grammatical errors is needed.

Title: It is not clear from the title that increasing water consumption is a focus in the manuscript.

Abstract: The abstract needs to be rewritten. It misrepresents the aims of the study, does not clear lay out methods, is not clear on significance of results, and has no conclusion statement.

Line 14: Placebo is not the appropriate word choice.

Keywords: I suggest replacing sugar with water and  children, behaviour change, and obesity with childhood obesity.

Introduction:

Lines 30-33: Include a citation.

Lines 43-45: This is not really set up by the previous paragraph where you seem to advocate for drinks sweetened with non nutritive sweeteners. 

Lines 84-85: You should clearly define and introduce health literacy.

Methods:

Table 1: I am not sure if breaking down the sample by gender is helpful as a table. Is gender vital somehow to the intervention?

Lines 115-116: Clarify whether participation was determined at the student or school level.

Lines 123-129 are a great justification for intervention and belong in the introduction. 

Lines 150-151: How accurate it this if you are asking adolescents to estimate serving size?

Lines 161-164: Provide a justification for these exclusions.

Table 2 is not necessary.

Explain in the methods section that you test for normality. This should not be in the results section.

Line 189: Did you use a t-test or the Mann-Whitney test?

Move up the intervention subsection to the beginning of the methods section and expand this section. After reading the intervention section, I do not have a clear understanding of what was implemented in schools. More detail is needed on the social marketing approach and the environmental changes in school. The environmental changes seem like they happened prior to the study. Is this the case? Did these changes take place in the comparison schools as well?

Line 210: Were focus groups conducted with primary school age youth while the intervention was conducted with youth ages 10-16? Why is there a discrepancy there?

Table 3 does not convey enough information to help readers understand the interventions. What channels were used for communications?

Results: Effect size and significance need to be reported throughout the results section. It is very difficult to judge if the interventions had any statistically significant or real world effects based on the way results are presented currently. Include descriptive statistics.

Table 4: Why did you not run an ANOVA to account for differences between groups at the first time point? What do the numbers in the category column stand for? This is not clearly defined.

Table 5: It is not clear if these changes are significant.

Table 6: You don't define A, B, and C. Aren't there four groups?

Discussion: It is not clear based on how the results were presented that these conclusions are logical. 

Line 324: How do nudges relate to your results? You mention nudges in the introduction and discussion, but do not say they are part of the intervention. Clarity on the interventions used would help this. 

Line 361: It was not clear the active intervention only lasted for 5 months. Where was this in the protocol?

Lines 365-367: It is a significant limitation if you lost more participants from the overweight or obese category.

Author Response

We appreciate the constructive suggestions and the opportunity to revise our paper, which we believe allowed us to develop a clearer and more compelling paper. We carefully considered and implemented a majority of  the reviewers’ comments and suggestions. Where revisions were not entirely possible, we have provided a response  with a rationale supporting our decision.

The revised manuscript was also proof-edited by a professional editing service. We believe that this too has considerably improved the flow of the article. However, this resulted in many changes in the manuscript besides those pointed by the reviewers. To improve the visibility of the main changes made, we accepted the tracked changes and highlighted only those parts in the text where considerable (content related) changes were made.

Please find a list of point-by-point responses to each comment and suggestion in the attached file. We hope that you find our revisions and responses sufficient and convincing.

Kind regards,

The authors

Reviewer 2 Report

The aim of the research was to find an effective way to replace the consumption of sweetened beverages with water in school children. It is not easy to understandd why the results of the study based on an important problem concerning youth development took so long to be published? Were the conditions the same 10 years ago, when the research was planned and started?

I have no objections to study design and analysis of results. The main finding that the mere change in the offer of a school shop without proper education is counterproductive, is interesting. The methodology section is overwhelmed by the addition of a section on the local relevance of research, which should rather be included in the introduction.

Author Response

We appreciate the constructive suggestions and the opportunity to revise our paper, which we believe allowed us to develop a clearer and more compelling paper.

The revised manuscript was proof-edited by a professional editing service. We believe that this too has considerably improved the flow of the article.

To improve the visibility of the main changes made, we highlighted only those parts in the text where considerable (content related) changes were made.

Please find below a list of point-by-point responses to each of your comment and suggestion. We hope that you find our revisions and responses sufficient and convincing.

Kind regards,

The authors

Round 2

Reviewer 1 Report

Thank you for the chance to re-review this manuscript. The changes you have made have significantly improved the paper. There are a few more modifications that should be considered.

Abstract:

In lines 11-12 change the language so you are not repeating "aimed at decreasing."

Are there clear definitions of SSBs, beverages with sugar and sweet beverages? These should be included.

Line 22: Add A before communication.

Introduction:

Lines 36-42: This is a good justification.

Are you already referring to your study  in line 86 when you say "This study"?

Line 123 is where I was referring to your work with primary schools. In some countries, this refers to younger children in grades k-5 (ages 5-10). Adding clarification of the ages included in the formative research and in line 159 will help to clarify.

Line 132 still refers to Table 3, though this should be Table 1.

Methods:

Line 176: I do not see an explanation of this in section 2.4. What is being explained there? Are you referring to information in table 3? Just clarify.

Line 191: Change was to were.

Lines 226-231: Adding a brief justification on these exclusions would be helpful.

Discussion:

Lines 388-389: This is true, but it is worth noting that collecting data the way you did is more likely to accurately reflect the true impact of your study. 

Lines 395-396: I think you should be explicit that dealing with perceived weight bias is an important consideration for future school-based interventions.

Author Response

Nutrients

Editors-in-Chief:

Prof. Dr. Maria Luz Fernandez and

Prof. Dr. Lluís Serra-Majem

March 8th, 2022

COVER LETTER

Manuscript ID: nutrients-1608522

Revised Manuscript: Water wins, communication matters: School-based intervention to reduce intake of sugar sweetened beverages and increase in-take of water”.

Dear Editors,

We appreciate the constructive suggestions and the opportunity to revise our paper. We carefully considered and implemented all of the reviewer’s comments and suggestions. Please find below a list of point-by-point responses to each comment. We hope that you find our revisions and responses sufficient and convincing.

Yours sincerely,

Prof. Nataša Fidler Mis, MSc, PhD

On behalf of all co-authors

Comments and suggestions for authors

 Response to REVIEWER 1 Comments, round 2

Point 1: Abstract: In lines 11-12 change the language so you are not repeating "aimed at decreasing."

Response 1: Thank you. We changed "aimed at decreasing" into “designed for reducing”.

 Point 2: Are there clear definitions of SSBs, beverages with sugar and sweet beverages? These should be included.

Response 2: Definition for beverages with sugar was already presented in lines 19 and 20 of abstract. It means all beverages that contain free sugars, which is SSBs and fruit juices.

We added definition for sweet beverages in abstract (lines 20 and 21) and in results (lines 273 and 274). It means beverages with sugar, low-calorie sweeteners and/or noncaloric sweeteners, such as sugar alcohols aspartame, acesulfame-K, saccharin, sucralose or stevia.

Point 3: Line 22: Add A before communication.

Response 3: We corrected “Communication” into “A communication”.

Point 4: Introduction: Lines 36-42: This is a good justification.

Response 4: Thank you for your comment.

Point 5: Are you already referring to your study  in line 86 when you say "This study"?

Response 5: Thank you for your comment. In line 86 we are not referring to our study, but to literature data. We corrected "this study" to “previous studies”.

Point 6: Line 123 is where I was referring to your work with primary schools. In some countries, this refers to younger children in grades k-5 (ages 5-10). Adding clarification of the ages included in the formative research and in line 159 will help to clarify.

Response 6: We focused on the grades 6-9 of primary-school in Slovenia. We corrected primary-school-age youths (i. e. grades 6–9 in Slovenia, i. e. aged 10–16 years).

Point 7: Line 132 still refers to Table 3, though this should be Table 1.

Response 7: Thank you. We corrected.

Point 8: Methods: Line 176: I do not see an explanation of this in section 2.4. What is being explained there? Are you referring to information in table 3? Just clarify.

Response 8: In the previous line 176, which is now line 177 we refer to section 2.4 Data collection and Table 3, where there is explanation about the terms “active” and passive” interventions. This is all correct.

In the line 209 there was a mistake, “Table 2”, which we corrected to “Table 3.”

Point 9: Line 191: Change was to were.

Response 9: Thank you for your comment. We corrected.

Point 10: Lines 226-231: Adding a brief justification on these exclusions would be helpful.

Response 10: We added a brief justification.

Point 11: Discussion: Lines 388-389: This is true, but it is worth noting that collecting data the way you did is more likely to accurately reflect the true impact of your study.

Response 11: Thank you for your comment. We agree and have added to the lines 391-939 the remark as you proposed.

Point 12: Lines 395-396: I think you should be explicit that dealing with perceived weight bias is an important consideration for future school-based interventions.

Response 12: We agree. We have added your remark to lines 400 and 401.
